# Coronal and Transverse Malalignment in Pediatric Patellofemoral Instability

**DOI:** 10.3390/jcm10143035

**Published:** 2021-07-08

**Authors:** Robert C. Palmer, David A. Podeszwa, Philip L. Wilson, Henry B. Ellis

**Affiliations:** 1Scottish Rite for Children, Dallas, TX 75219, USA; palmrc5@gmail.com (R.C.P.); David.Podeszwa@tsrh.org (D.A.P.); Philip.Wilson@tsrh.org (P.L.W.); 2Department of Orthopeadics, University of Texas Southwestern Medical Center, Dallas, TX 75033, USA

**Keywords:** pediatric patellar instability, coronal malalignment, genu valgum, rotational malalignment, femoral anteversion, tibial torsion

## Abstract

Patellofemoral instability (PFI) encompasses symptomatic patellar instability, patella subluxations, and frank dislocations. Previous studies have estimated the incidence of acute patellar dislocation at 43 per 100,000 children younger than age 16 years. The medial patellofemoral ligament (MPFL) complex is a static soft tissue constraint that stabilizes the patellofemoral joint serving as a checkrein to prevent lateral displacement. The causes of PFI are multifactorial and not attributed solely to anatomic features within the knee joint proper. Specific anatomic features to consider include patella alta, increased tibial tubercle–trochlear groove distance, genu valgum, external tibial torsion, femoral anteversion, and ligamentous laxity. The purpose of this paper is to provide a review of the evaluation of PFI in the pediatric and adolescent patient with a specific focus on the contributions of coronal and transverse plane deformities. Moreover, a framework will be provided for the incorporation of bony procedures to address these issues.

## 1. Introduction

Patellofemoral instability (PFI) encompasses symptomatic patellar instability, patella subluxations, and frank dislocations. Dislocations are almost always to the lateral side of the femoral trochlea. Patellar subluxation occurs when the patella partially dislocates but does not fully dislocate out of the trochlear groove [1]. In order to standardize terminology, Parikh and Lykissas provided a comprehensive 4-part classification for lateral PFI that includes first-time dislocators (type I), recurrent patellar dislocation (type II), dislocatable patella (type III), and dislocated patella (type IV) which have further sub-classifications for each type [2]. Frosch and Schmeling have published a classification scheme with similar considerations [3].

The incidence of acute patellar dislocation has been estimated at 43 per 100,000 children younger than age 16 years [4]. In a pediatric population, greater than 60% of those with a first-time dislocation may go on to recurrence, which is reported more commonly in females (~70%) than males [5,6]. A history of recurrent instability (two or more episodes) is predictive of future instability [7]. Younger children (<14 years of age) and those with trochlear dysplasia are more likely to experience recurrent dislocations [8,9]. *Hevesi* et al. developed a scoring system known as the recurrent instability of the patella score that focuses on four factors, including age <25 years (2 points), skeletal immaturity (1 pt), trochlear dysplasia (1 pt), and the tibial tubercle–trochlear groove to patellar length ratio (TT–TG/PL) (1 pt). Patients were stratified into three groups of low- (0–1), intermediate- (2–3), and high-risk (4–5) groups based on these factors. Instability-free survival was provided at 1, 2, 5 and 10 years; for the high-risk group, this was 84.4% at 1 year, 62.5% at 2 years, 34.4% at 5 years, and 20.8% at 10 years [10].

The medial patellofemoral ligament (MPFL) complex is a static soft tissue constraint that stabilizes the patellofemoral joint serving as a checkrein to prevent lateral displacement. Other static osseous constraints include trochlear and patellar morphology and skeletal alignment, which should be evaluated in both the coronal (angular) and transverse (rotational) planes with regard to PFI [11]. Specific anatomic features to consider include patella alta, increased tibial tubercle–trochlear groove distance, genu valgum, external tibial torsion, femoral anteversion, and ligamentous laxity [5,12,13]. The combination of excessive femoral anteversion, patella alta, an increased Q-angle, and excessive external tibial torsion is known as “miserable malignment” which has historically been associated with recurrent PFI [14,15].

As highlighted above, the causes of PFI are multifactorial and not attributed solely to anatomic features within the knee joint proper. Rarely, medial patellofemoral instability can occur and typically associated with an iatrogenic etiology (i.e., prior lateral release). Thus, the current manuscript will exclusively focus on lateral patellofemoral instability when referencing PFI. The purpose of this paper is to provide a review of the evaluation of PFI in the pediatric and adolescent patient with a specific focus on the contributions of coronal and transverse plane deformities. Moreover, a framework will be provided for the incorporation of bony procedures to address these issues.

### 1.1. Patient History

Depending on the type of PFI, the patient may describe the symptoms in many different manners. Some may merely describe frequent falling, while many endorse fear of a ‘knee cap’ problem with no true dislocation. In the acute, traumatic patellar dislocation, the patient may describe hearing or feeling a pop at the time of injury with observation of a knee deformity and swelling when the patella dislocates laterally. Specific points of the patient’s history to elicit should include the patient’s level of activity, mechanism of the event, history of previous dislocations, total number of dislocations, how the dislocation reduced, and the timing and severity of effusion following the event. Dislocations commonly spontaneously reduce or reduce with extension of the knee, but a minority may require assisted reduction with sedation. Acutely following instability events, the physician should carefully assess the patient for mechanical symptoms which may suggest the presence of a loose body from an osteochondral injury. An understanding of the patient’s activity level can also be valuable at the onset as this can help guide treatment decision making.

Emphasis is also placed on obtaining a thorough family history of PFI, as well as ligamentous laxity and associated conditions (e.g., generalized ligamentous laxity, Ehlers–Danlos syndrome, Marfan Syndome, Down syndrome, Ellis–van Creveld syndrome, nail-patella syndrome, Rubenstein–Taybi syndrome, Kabuki syndrome, hypotonic cerebral palsy, and hypoplastic patella syndromes), which is critical to evaluate [1].

### 1.2. Physical Exam

A standard physical examination of PFI should begin with evaluating the patient’s standing limb alignment including an assessment of genu valgum and varum. Subtle deformity may be difficult to grossly visualize if not suspected. Significant genu valgum deformity can be assessed during an evaluation of gait or by having the patient stand upright. Genu valgum can be quantified by the intermalleolar distance. If significant femoral anteversion is present, one will appreciate the “squinting/kissing patella” where the patient’s feet are facing forward and the patella is facing towards the midline as if to ‘kiss’ the contralateral patella (Figure 1) [16]. With special attention to rotational profile, upon gait assessment the examiner must document the patient’s foot progression angle (mean 10° external; range 3° internal to 20° external) [17]. The amount of generalized hyperlaxity may be an important consideration in evaluation of a patient with PFI. The Beighton hypermobility score is helpful in assessing ligamentous laxity (Table 1) [18,19]. A score of five or greater is indicative of a hypermobile condition and may warrant further evaluation and consideration in treatment.

With the patient sitting, active knee flexion and extension are examined to evaluate the patellar tracking and determine the presence of a J-sign. The patient sits on the edge of the table and the knee is allowed to flex from full extension. The examiner can hold the limb in terminal extension to allow for the quadriceps to relax in order to visualize the resting position of the patella before the patient begins to actively flex. As the knee begins to flex, the examiner releases enough support to stimulate the patient to contract the quadriceps. In the patient with a J-sign, the lateralized patella shifts medially just as the knee begins to flex. [1,20]. The J-sign can be classified as not present, mild, or severe. Several anatomic factors are contributory to the development of a J-sign; including rotational malalignment, patella alta and trochlear dysplasia.

Next, with the patient laying in a supine position with the quadriceps relaxed, patellar glide is quantified by placing a medial translation force followed by a lateral one, using the width of the patella divided into quadrants as reference based on percentage of the patella translated (25%, 50%, 75%, 100% [dislocates]); in addition to this, the patellar endpoint, whether firm or soft, is recorded. The Fairbank patellar apprehension sign is evaluated by placing a laterally directed force on the patella with the knee in 30° of flexion and is positive if the patient indicates discomfort or apprehension [21]. The patellar tilt is evaluated with the knee fully extended and the quadriceps relaxed. The examiner attempts to lift the lateral border of a tilted patella which should correct to at least neutral; if not, this suggest tightness of the lateral structures. [22]. Historically, the Q-angle—measured with the patient in the supine position as the angle between the ASIS and patella and patella and tibial tubercle—has been shown to be increased in patients with PFI [23,24]. Reference range values are dependent upon the patient’s sex as well as positioning: supine male 8–16°, supine female 15–19°, prone male 11–20° and prone female 15–23° [25].

Femoral and tibial rotation are evaluated in the prone position with the hip extended as it best simulates the patient’s hip position during gait and stabilizes the pelvis [16,26] (Figure 2). While in this position with the knees flexed, both hip external and internal rotation are quantified. *Staheli* et al. reviewed 500 lower-extremity rotational profiles and provided reference values: hip internal rotation for males 50° (25–65°) and for females 40° (15–60°); hip external rotation 45° (25–65°) [17].

The Craig’s test can be used to quantify the patient’s femoral anteversion by utilizing the greater trochanter as a reference landmark. The patient is placed prone with the knee flexed to 90, the greater trochanter is palpated and the hip rotated until the greater trochanter is felt to be parallel to the examination table, the angle the leg makes with a vertical orthogonal line from the table in this position is the patient’s anteversion [27,28]. The accuracy of this test has been debated when compared to imaging [29,30]. Normative values for femoral anteversion range from 7° to 20° [31]. The thigh foot angle is an assessment of tibial torsion. It is measured in the prone position with the knees flexed 90 degrees and is angle between the long axis of the thigh and the long axis of the ipsilateral foot. Alternatively, the transmalleolar angle is measured with the patient supine with the patella pointing straight up. The transmalleolar angle is the angle formed between the transmalleolar axis (line drawn between the lateral and medial malleoli) and the plane of the floor. The average thigh foot axis measures 10° external rotation (range 5° internal to 30° external) and the transmalleolar angle is 20° external (range 0° to 45° external) [17] (Figure 3). Tamari et al. have acknowledged the limitations in clinical evaluation when compared to imaging techniques [32].

## 2. Imaging

In the setting of an acute injury, preliminary radiographic evaluation should include at least three views of the affected knee (anterior-posterior, lateral, and patellar views). Various types of patellar views have been described, but bilateral Merchant or Laurin (20–30° knee flexion) views have the most value in the evaluation of PFI [33,34]. The utility of patellar radiographs in these lesser degrees of flexion is to visualize the patella in the position in which it engages with the trochlear groove. For a patient presenting with chronic complaints of instability, routine imaging in our practice includes an anterior posterior (AP) view of the affected knee as well as a true lateral view with the knee flexed at 30° (to allow for assessment of trochlear dysplasia and patellar height), a merchant view (with the quadriceps muscle relaxed to allow for assessment of passive patellofemoral alignment), and an AP bilateral lower-extremity standing alignment film with the patella forward when coronal malalignment is suspected or in a skeletally immature patient.

Patella height can be measured using the Caton–Dechamps index—measured on a lateral knee radiograph as the distance from the inferior aspect of the patellar articular surface to the anterior aspect of the tibial plateau divided by the length of the patellar articular surface—which has been shown to be a more reliable method for skeletally immature patients [35,36]. The lateral radiograph is used to qualify the patient’s trochlear dysplasia utilizing the Dejour classification [37]. This classification is composed of 4 categories describing the increased severity of trochlear dysplasia (Table 2). A recent study questioned the reproducibility of the Dejour classification and offered that a revised MRI classification may be more reliable [38]. The MRI Dejour classification utilizes axial MRI imaging to evaluate trochlear dysplasia and retains the original classification groups (A-shallow trochlea >145°, B-flat trochlea, C-lateral convexity medial hypoplasia, and D-cliff).

The presence of open physes and the patient’s estimated growth remaining based on bone age are important factors when considering medial patellofemoral ligament reconstruction or guided growth to correct limb malalignment. In order to make this determination, a left AP hand film can be utilized to determine a bone age [39]. Traditionally, coronal plane alignment has been evaluated with an orthoroentgenogram (3-foot standing AP lower-extremity alignment radiograph) or a teleoroentgenogram (a single-exposure weightbearing study) versus obtaining three separate standing AP images that were then stitched together. More recently, these modalities have been supplanted by using low-dose radiation biplanar fluoroscopy in order to minimize radiation exposure (EOS Imaging, Paris) [40]. Another advantage of biplanar fluoroscopy is the ability to measure absolute leg lengths as there is no magnification error, though it is important to note that patient movement during the capture may create measurement inaccuracies.

Using the standing radiograph, the mechanical axis of the lower extremity is determined by drawing a line from the center of the femoral head to the center of the tibio-talus mortise. The knee is divided into zones based upon the distal femur as follows: intercondylar is neutral, a line bisecting the condyle denotes zone 1 and 2, and zone 3 is defined as falling outside the margin of the femoral epicondyle, indicating increasing genu valgum and negative values worsening genu varum (Figure 4). The mechanical axis deviation is another measurement to quantify coronal plane alignment and is the distance between the center of the knee (intercondylar femoral notch) and the patient’s mechanical axis line [41,42]. Additional measurement considerations are the mechanical lateral distal femoral (mLDFA) and mechanical medial proximal tibial (mMPTA) angles, which reference the mechanical axis of each bone segment. The mLDFA is the lateral angle formed by the line from the center of the femoral head to the femoral notch and the tangential line of the femoral condyles. The mMPTA is the medial angle formed by the tangential line of the tibial plateau and the long axis of the tibia. These angles are measured in order to determine whether the distal femur, proximal tibia, or a combination thereof is contributing to the patient’s coronal plane deformity. The standard values mLDFA and mMPTA are both 87° (85–90°) [41,42,43]. It is critical that the standing film is performed with the knees in full extension and the patella forward, as knee flexion or rotation for any reason will result in inaccurate measurements [44].

Routine use of magnetic resonance imaging (MRI) is surgeon or institution dependent. An MRI of the knee without contrast may be indicated for an isolated traumatic patellar dislocation with a large effusion to evaluate for an osteochondral or chondral injury, recurrent PFI with mechanical symptoms, or to aide with treatment recommendation. The MRI can also provide a rotational assessment including femoral version or tibial torsion as described below and should be considered when obtaining an MRI of the knee.

CT has become an increasingly popular technique for assessing rotational profile given its reliability and reproducibility. It should be noted, however, measurements are technique dependent and increased radiation exposure is a consideration especially in the pediatric population [40,45,46,47,48,49]. Kaiser et al. compared several techniques of measuring femoral and tibial torsion (Waidelich, Murphy, Yohshioka, Hernandez, and Jarrett techniques). They demonstrated comparable mean values to previously published values for each technique but showed that a measurement by the Hernandez technique could represent a pathologic torsion value while being within anatomic reference when utilizing the Waidelich technique [50,51,52]. A subsequent study by *Schmarazner* et al. used CT scan to compare five measurement techniques which evaluated the location of the femoral neck axis in a proximal to distal fashion and found the most pronounced difference between the Lee (most proximal) and the Murphy (most distal) techniques; all techniques had excellent agreement for intraobserver (ICC, 0.905–0.973) and interobserver reliability (ICC 0.938–0.969) [53]. Thus, it is paramount to familiarize with the technique at one’s institution and the respective reference values for that specific technique. Consistency within an institution or a practice is paramount as these techniques do have variation especially in patients with significant dysplasia.

At our institution, we utilize the Jarrett method for CT scans given the accuracy of this technique as well as its anatomic basis. An axial oblique image is necessary for this technique. A line is drawn on a single axial oblique image that runs from the center of the femoral head through the center of the femoral neck (Figure 5) [54]. The angle is measured with a tangential line through the distal femoral condyles.

Alternatively, Rosskopf et al. demonstrated that low-dose biplanar radiographs are reliable when compared to CT, and subsequently MRI, when obtaining a lower-extremity rotational profile [55,56]. Limb-alignment MRI protocols have been developed which have decreased exam time requirements, though the relative cost of an MRI remains a significant consideration in the US healthcare system [46,57]. Sung et al. developed and validated a mobile application that can reliably measure femoral anteversion from AP and lateral femur radiographs [58]. In an MRI study comparing a population of patients with recurrent patellar dislocations and controls, Maine et al. quantified the rotational alignment of the extensor mechanism, known as the quadriceps torsion angle (QTA). This measurement was shown to be reliable and reproducible and in the setting of increased femoral anteversion was an additive risk factor for recurrent patellar dislocation [59].

Tibial torsion is defined as the physiologic rotation of the tibia from the proximal to the distal articular axis of the tibia in the transverse plane and historical attempts have been made to measure this angle using both radiographic and later CT imaging [49,60]. An earlier method proposed by Jend et al. used a CT scan and measured the posterior tibial condylar axis just above the fibular head proximally and the tibial pilon angle just above the talocrural space distally. This is the method that is utilized at our institution (Figure 6) [61]. A recent study by Liodakis et al. evaluated the Ulm, Jend and bimalleolar methods for tibial torsion and all three methods demonstrated excellent ICC scores (Ulm 0.918, Jend 0.916, bimalleolar axis 0.92). Notably, on average both the Ulm and Jend methods underestimated the bimalleolar axis measurements by 4.8° and 13°, whereas the Jend overestimated the Ulm by 8° [62]. Again, as discussed previously, these findings lend support to institution-wide standardization given the variability among measurement techniques.

A lateralized patellar tendon insertion has been implicated as a risk factor for lateral PFI and has been quantified by measuring the distance between the midline insertion of the patellar tendon onto tibial tubercle and the center of the trochlear groove measured on the superior-most axial cut exhibiting full cartilage coverage of the posterior femoral condyles (Figure 7) [63,64]. *Bernhold* et al. found that the TT–TG could be measured in 82% of their radiographic patellofemoral view study cohort and that this measured 5–8 mm smaller than MRI TT–TG. Studies have noted consistently smaller TT–TG values by MRI (approximately 4 mm) compared with CT [65,66,67,68]. Dickens et al. determined the mean value for TT–TG in a pediatric population using 3-T MRI to be 8.6 mm in the control group and 12.2 mm for the comparison group with PFI [69].

*Seitlinger* et al. proposed an alternative method for quantifying a lateralized patellar tendon insertion utilizing the tibial tubercle-posterior cruciate ligament distance, which is measured from the midpoint of the insertion of the patellar tendon to the medial border of the posterior cruciate ligament (Figure 8) [70]. The distal tibial condylar line (dTCL) is a tangential line of the proximal tibia that is distal to the articular surface and proximal to the fibular head. The proposed advantages of this measurement is it overcomes the difficulty in measuring the deepest point of a dysplastic trochlea, does not vary as a function of knee flexion, and isolates the location of dysplasia. In contrast, the more commonly utilized TT–TG does not provide specificity regarding the relative contributions of tibial tubercle lateralization, medialization of the trochlear groove, and/or soft tissue malrotation through the knee joint.

More recently, increased tibial tubercle torsion has been highlighted as a risk factor for PFI. This rotational angle is measured from the posterior femoral condyles to the center of the tibial tubercle in a craniocaudal axis. Tibial tubercle torsion was significantly increased in the patient group with PFI (17.9° +/− 7.0°) when compared to a control group (5.8° +/− 3.6°) and correlated with the TT–TG measurement (r = 0.87) [71].

## 3. Treatment

Recognition and treatment of PFI have undergone considerable evolution in recent years, with more than 100 different surgical techniques reported in the literature [72,73,74,75,76]. In patients with a first-time dislocation without significant risk factors for recurrence and no osteochondral fracture, non-operative management is recommended [9]. In our practice, this consists of a short period of brace immobilization in full extension (3 weeks) and physical therapy as early mobilization (<3 weeks) has shown to increase the repeat dislocation rate three-fold [77]. Following early immobilization, physical therapy with a lateral buttress brace is used. Several recent surgical algorithms have been developed which generally focus on the patient’s skeletal maturity, TT–TG, trochlear dysplasia, coronal and rotational malalignment, and the presence of chondral lesions [78,79,80,81]. Medial patellofemoral ligament reconstruction has become the preferred method to address PFI and has demonstrated excellent results [82,83,84,85,86,87,88]. However, when coronal or transverse plane malalignment is present, consideration for correction of malalignment should be considered prior to MPFL reconstruction.

### 3.1. Coronal Plane Malalignment

Once coronal plane malalignment has been identified, several treatment options are available. Two primary factors for determining the appropriate corrective procedure include growth remaining determined by bone age and the deformity location. In patients with open physes, growth modulation is a treatment option with a relatively low morbidity and low complication rates for addressing PFI in patients with genu valgum [89]. Tension band plates about the physis are one treatment option. An important consideration is medial plates may necessitate staged treatment to avoid plate/graft impingement; while screw hemiepiphysiodesis may merit further study as an option that may be employed concurrent with MPFL [90,91]. No matter the technique, patient should be cautioned with the concepts of overcorrection and rebound which had been reported in both techniques. [92,93].

In skeletally mature patients, both opening and closing wedge osteotomies about the distal femur are reliable treatment options for coronal plane malalignment. In a retrospective cohort study of 18 patients (20 knees) with genu valgum and PFI, Frings et al. demonstrated that the utilization of a closing wedge distal femoral osteotomy in combination with MPFL reconstruction and tibial tubercle osteotomy (TTO) eliminated recurrence of re-dislocation with median follow up of 16 (12–64) months [94]. In a skeletally mature adolescent population with PFI and genu valgum (≥zone II) mechanical axis, Wilson et al. demonstrated that 80% of patients had no further episodes of instability following an isolated open wedge distal femoral osteotomy with mean correction of 10.4° (7° to 12°) at mean follow up of 4.25 years (range 3.2 to 6) [95]. Thus, when faced with PFI with genu valgum, coronal plane correction can add benefit even without a soft tissue procedure. For Zone II values or mLDFA <83 degrees, we recommend guided growth with transphyseal screw placement if skeletally immature with >9 months of estimate growth or a distal femoral open wedge varus producing osteotomy in the mature patient (see Figure 9). Addressing coronal plane malalignment has more benefit than just preventing recurrence of PFI as a central mechanical axis about the knee is optimal for long term joint preservation. Thus, when faced with genu valgum and PFI, we address genu valgum as the first stage of treatment for PFI without an MPFL. A staged MPFL is often times considered based on patient symptoms, athletic activity, and risk factors (Figure 9, Figure 10 and Figure 11).

### 3.2. Transverse Plane Malalignment

Transverse plane deformities have been recognized by previous studies as a potential driver of PFI and anterior knee pain with historical interest in femoral torsion as a potential driver of developmental dysplasia and osteoarthritis of the hip [15,16,96,97,98,99,100,101,102]. There are several methods for correcting excessive femoral anteversion with good clinical results and patient satisfaction, though it is important to be aware of the advantages and disadvantages of each method and its appropriate application to individual patient pathology [103]. Osteotomies can be placed about the intertrochanteric, subtrochanteric, diaphyseal, or distal metaphyseal regions and fixed with a plate, an intramedullary nail, external fixator, or a combination thereof [43,104,105]. The literature has described corrective osteotomy in patients with radiographic femoral anteversion >20° to 25° [106,107,108,109]

The effect of transverse plane correction on other planes is an important consideration [110,111]. Using computer modeling from CT data of a femur, Nelitz et al. demonstrated that proximal femoral osteotomies created varus alignment and distal osteotomies created valgus alignment [112]. In a cadaveric model, Kaiser et al. demonstrated that femoral derotational osteotomy has a significant impact on patellar tilt and axial plane engagement with a modest change in the TT–TG distance [113]. Several studies have evaluated the use of 3D-printed cutting guides both in cadaveric specimens and in vivo with promising results which may serve to provide precise and reproducible osteotomy correction in multiple planes with less radiation exposure to patients [114,115,116,117,118,119]. Our preferred technique is a midshaft femoral derotational osteotomy over an antegrade intramedullary nail. Though historical literature has reported the occurrence of fat embolism syndrome in the pediatric patient and deformity correction, in our practice, by utilizing the previously reported technique of femoral venting prior to reaming, we have not experienced this complication (see Figure 12) [120,121,122,123,124,125,126,127]. Another advantage of this intramedullary technique is that the reamings serve as bone graft at the osteotomy site. In the setting of a 1 cm incision over the osteotomy site and a low-energy transverse corticotomy, the limited disruption of the soft tissue envelope minimizes periosteal disruption. Thus, it is expected that the non-union rate would be exceedingly low in a healthy adolescent patient with a load sharing device and the added benefit of early weight bearing. However, Teitge has reported complications in the adult population with intramedullary nailing which ranging from iatrogenic fracture, increased postoperative pain and blood loss, delayed union and death from fat embolism, and as such, has transitioned to using a proximal femoral osteotomy stabilized by a 95-degree condylar blade plate [120].

Krengel and Staheli reported the results of 52 tibial rotational osteotomies (39 proximal, 13 distal tibia) and reported five (13%) serious complications with proximal tibial osteotomies including compartment syndrome, peroneal nerve palsy and deep infection while there were no significant complications in the distal osteotomy group [128]. Delgado et al. reported the results of 13 rotational osteotomies for femoral anteversion, tibial torsion, or both that all healed without complication with clinical and radiographic improvement [129]. In a study using various correction techniques in a patient population with miserable malignment syndrome, Bruce and Stephens did not find a significant difference in results with regard to the level at which the tibial osteotomy was performed. The authors also mention that a concomitant fibular osteotomy was not performed unless tibial rotational correction was greater than 35° [15]. As peroneal nerve palsy and compartment syndrome are known complications of proximal tibial osteotomies, some authors have recommended prophylactic peroneal nerve decompression and anterior compartment fasciotomy. Some authors advocate a prophylactic peroneal nerve decompression when acute correction greater than 5° of varus or valgus or greater than 45° of rotational correction is performed [44,130,131,132,133]. However, in the authors’ experience, this is not usually considered until there is greater than 10° of coronal plane correction and there is certain discretion given to the location of the osteotomy. In a study of tibial torsion in an adult population, Turner demonstrated a correlation of PFI in patients with increased external tibial torsion [134] *Staheli* indicated surgical intervention for patients older than 8 years of age with tibial torsion greater than 30° [135].

The author’s preferred technique is the utilization of distal tibia and fibula osteotomy with gradual correction using a circular external fixator with surgeon discretion with regard to implementing acute correction at the time of surgery. Intraoperative neuromonitoring is used at our institution which we have found allows for safe acute intraoperative correction. Additional benefits of gradual correction include early weight bearing, no residual implants and the ability for shared decision making with the patient and family to determine final rotational alignment. Although the authors consider rotational malalignment when treating recurrent patellofemoral instability, a medial patellofemoral ligament reconstruction is often times first-line treatment even with significant rotational malalignment. Rotational osteotomies are considered in combination with a MPFL reconstruction in a revision PFI with severe deformity (femoral version >40 degrees, tibial external torsion >35 degrees, severe J-sign/lateral tracking, and/or high-grade trochlear dysplasia) (Figure 12).

## 4. Conclusions

The importance of limb alignment and rotational profile considerations in the treatment of PFI cannot be overstated. Eckhoff wisely surmised, “The static and dynamic relationships of the underlying tibia and femur determine the patellar tracking pattern.” [98]. Thus, a comprehensive clinical evaluation and appropriate consistent imaging modalities are indicated to appropriately identify and address a patient’s underlying pathology. Several options exist for correction and stabilization of coronal and transverse plane deformities, and it is essential that the surgeon be familiar with the advantages and disadvantages of each technique. In doing so, they may employ the optimal technique for patient’s unique deformity in order to maximize the patient’s outcome.

## Figures and Tables

**Figure 1 jcm-10-03035-f001:**
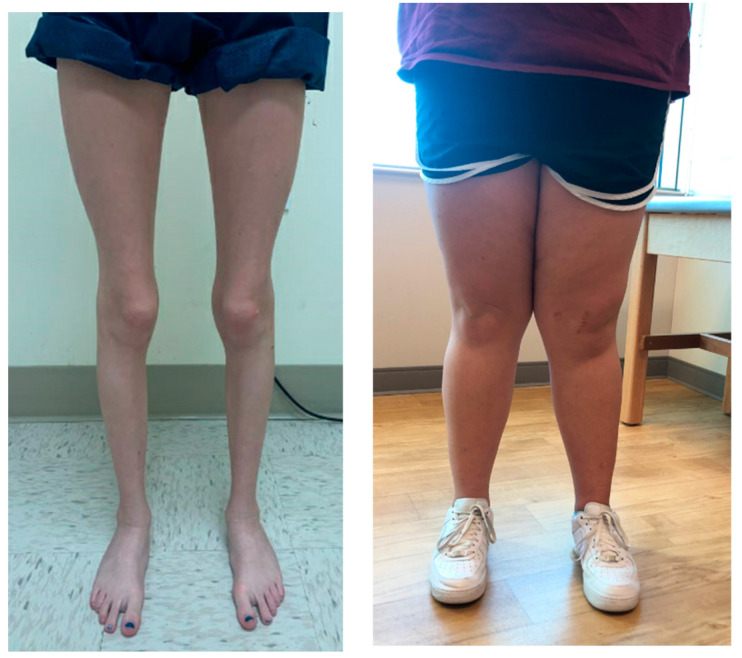
((**left**) image) Clinical appearance of excessive femoral version in a girl. With the knees in full extension and the feet aligned (pointing straight forward), the patellae face inward. ((**right**) image) Another patient demonstrates significant outward tibial angulation with the patella facing forward, indicating significant clinical genu valgum.

**Figure 2 jcm-10-03035-f002:**
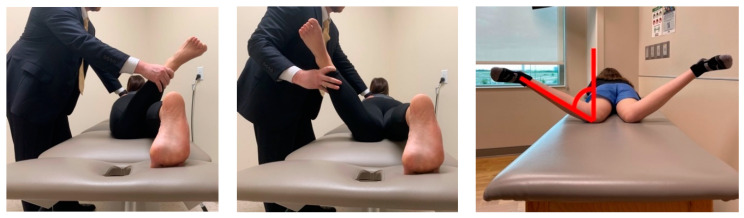
Both hip external (**left**) and internal (**middle**) rotation are assessed with the patient prone, and the knee flexed to 90°. A vertical line is utilized for reference as demonstrated (**right**) for determination of the internal rotation angle.

**Figure 3 jcm-10-03035-f003:**
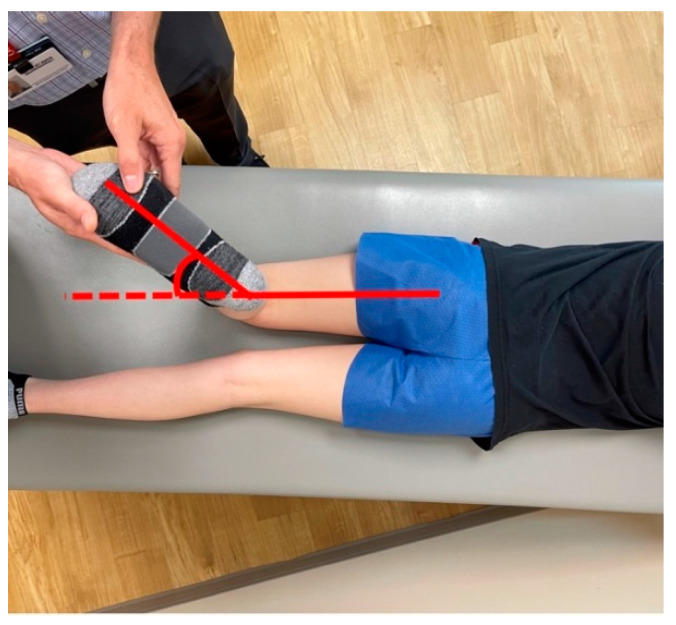
Tibial torsional profile examination with the patient prone. The examiner can assess the thigh–foot axis to estimate tibial torsion.

**Figure 4 jcm-10-03035-f004:**
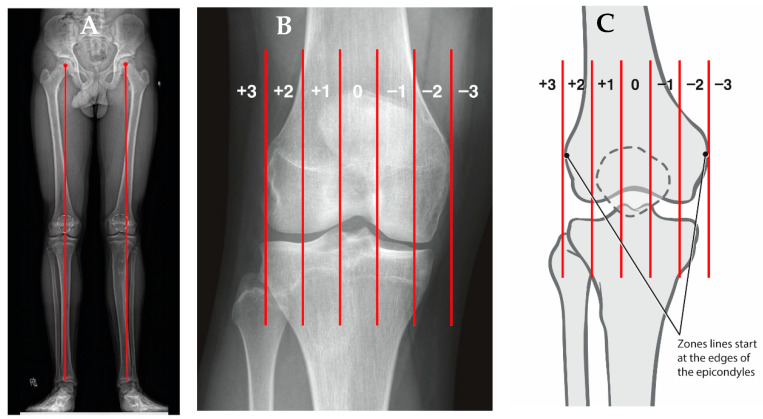
(**A**) AP standing alignment films demonstrates the patient’s mechanical axis is lateral to the center of the knee indicating genu valgum. (**B**) The knee is divided into zones based upon the distal femur as follows: intercondylar is neutral, a line bisecting the condyle denotes zone 1 and 2, and zone 3 is defined as falling outside the margin of the femoral epicondyle, indicating increasing genu valgum and negative values worsening genu varum (**C**).

**Figure 5 jcm-10-03035-f005:**
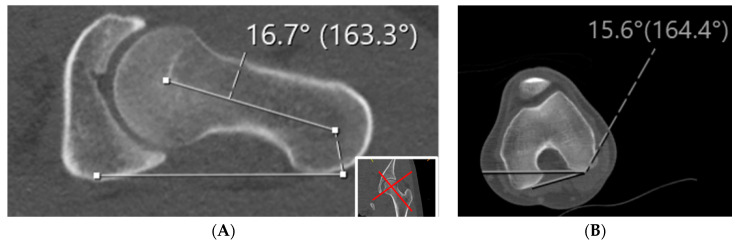
The Jarrett methods for measuring femoral version is demonstrated in the CT scan. As compared to other methods, the Jarrett method uses the axial oblique down the femoral neck to measure the proximal aspect of the femur. (**A**) Axial oblique as defined by the images in the bottom right corner to measure the angle perpendicular to the patient lying flat on the table with the line created down the femoral neck. This value is then added to angle measured in (**B**) in which the angle created by the femoral condyles is measured.

**Figure 6 jcm-10-03035-f006:**
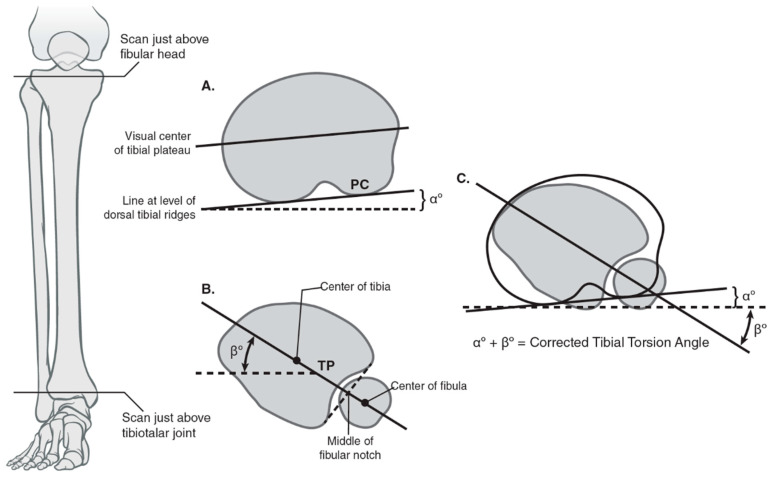
Calculating tibial torsion using posterior condylar angle and tibial pilon angle using the Jend method. (**A**) The coronal and axial planes used for measuring tibial torsion proximally with a line tangential to the posterior tibial condyles using the Jend method; the angle formed with the horizontal is denoted as α. (**B**) The coronal and axial planes used for measuring tibial torsion distal with a line formed by the center of the fibular notch and the center of the tibial pilon; the angle formed with the horizontal is denoted as ß. The best circle fit of the pilon includes the fibular notch but excludes the medial malleolus. (**C**) The summation of the α and ß angles provides the corrected tibial torsion angle.

**Figure 7 jcm-10-03035-f007:**
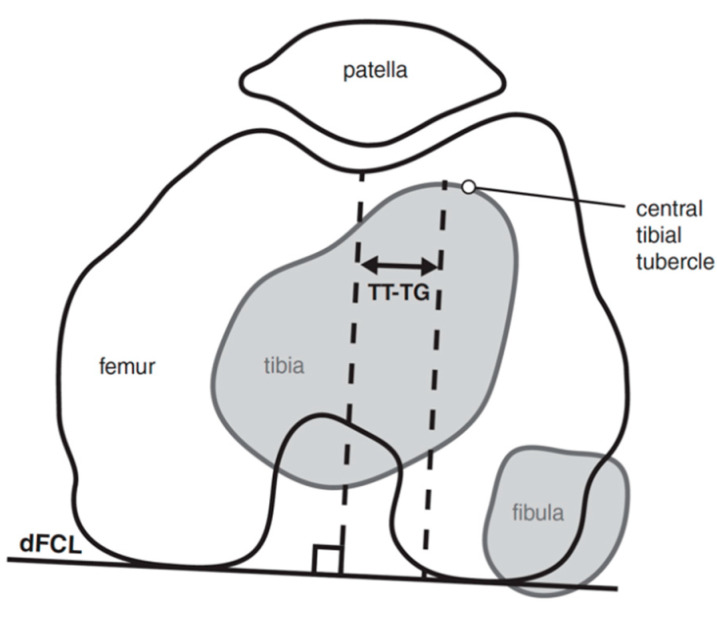
TT–TG measurement quantifies the distance from the midline insertion of the patellar tendon onto the tibial tubercle to the center of the trochlear groove measured on the superior-most axial cut exhibiting full cartilage coverage of the posterior femoral condyles.

**Figure 8 jcm-10-03035-f008:**
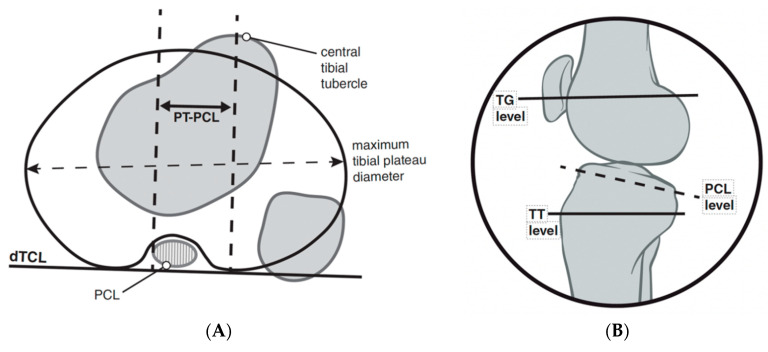
TT-PCL measurement quantifies the distance from the midpoint of the insertion of the patellar tendon to the medial border of the posterior cruciate ligament (**A**). The distal tibial condylar line (dTCL) is a tangential line of the proximal tibia that is distal to the articular surface and proximal to the fibular head that provides a linear reference line to measure the distance between perpendicular lines of the references points listed above. A sagittal figure (**B**) is used to demonstrate the location of the axial plane in which the medial aspect of the PCL should be used for this measurement.

**Figure 9 jcm-10-03035-f009:**
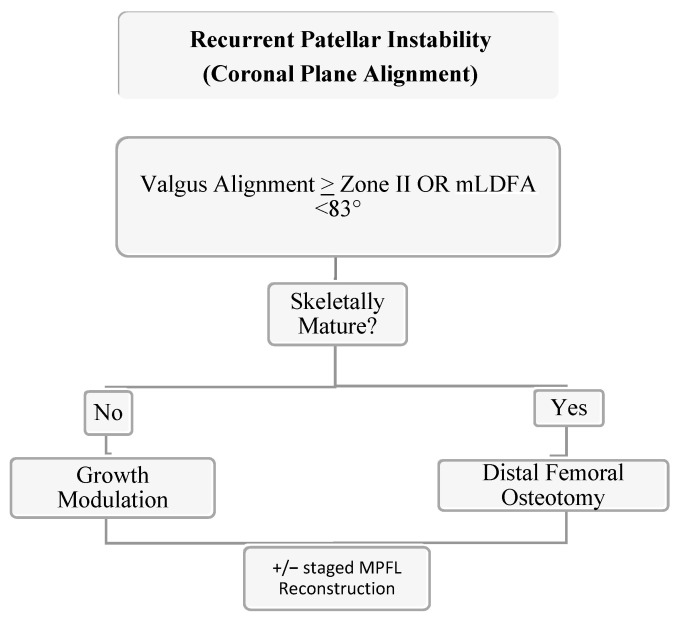
Authors’ preferred treatment algorithm for genu valgum and PFI.

**Figure 10 jcm-10-03035-f010:**
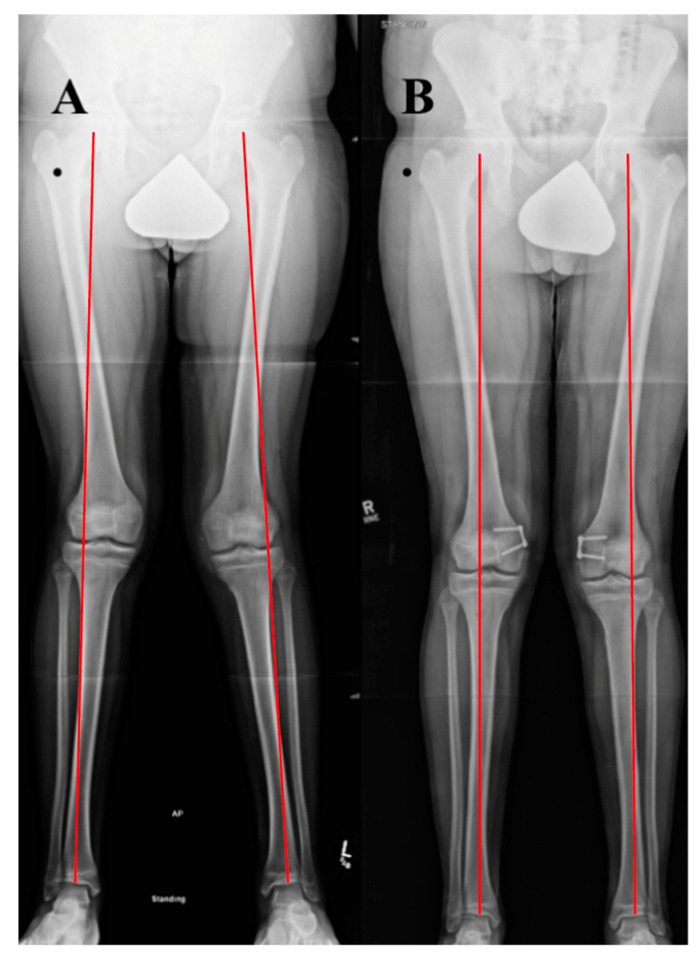
Case example of growth modulation. A 14-year-old male with right patellar instability who underwent bilateral guided growth (hemiepiphyseodesis) for genu valgum using a plate and screw construct over the medial distal femoral epiphysis. (**A**) A standing alignment radiographic preoperatively with Grade III bilateral genu valgum. (**B**) A 6 month postoperative standing alignment demonstrating correction of coronal plane malalignment.

**Figure 11 jcm-10-03035-f011:**
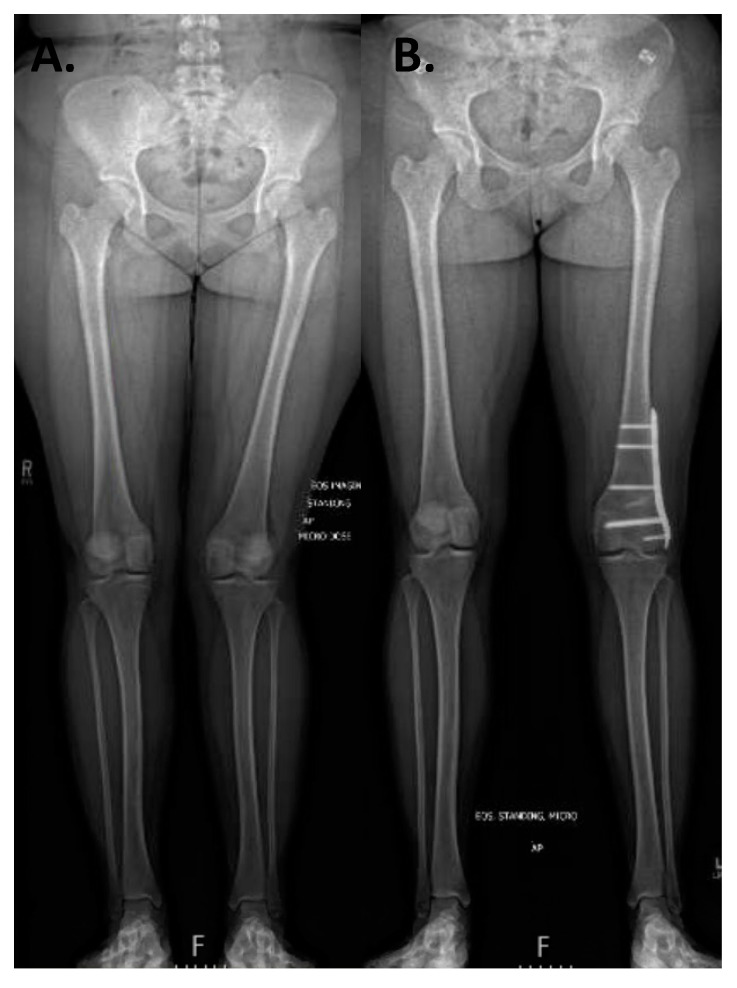
Case example of distal femoral osteotomy. A 16-year-old female with left genu valgum and recurrent patellar instability. (**A**) Preoperative standing alignment radiographs with asymmetric left valgus. (**B**) Postoperative images following a varus producing distal femoral osteotomy.

**Figure 12 jcm-10-03035-f012:**
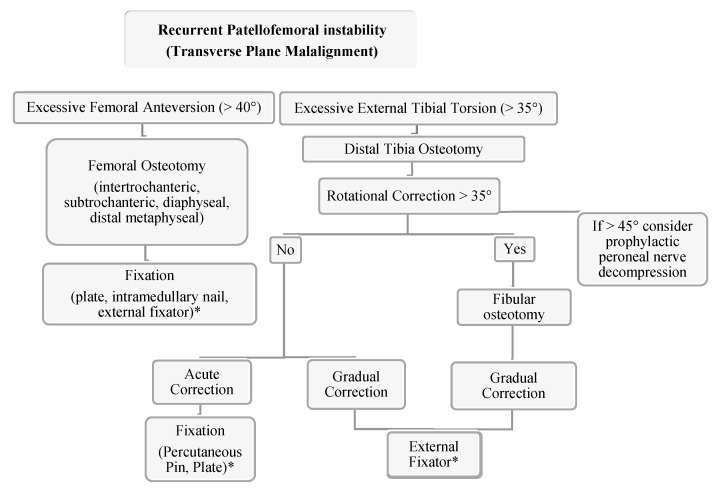
Authors’ preferred treatment algorithm for transverse plane malalignment with PFI. * The authors recommend the consideration of a medial patellofemoral ligament reconstruction for these cases.

**Table 1 jcm-10-03035-t001:** Beighton hypermobility score.

Physical Exam Finding	Points (1 Point for Each Side, 9 Total)
Knee hyperextension (>10 degrees)	2
Elbow hyperextension (>10 degrees)	2
Metacarpophalangeal joint extension >90°	2
Ability to flex thumb to forearm	2
Place palms flat on floor on forward bend	1

**Table 2 jcm-10-03035-t002:** Lateral imaging findings characteristic of the Dejour classification.

Dejour Type	Lateral Radiograph Findings	Significance
Type A	Crossing sign	Shallow trochlea; trochlear groove lies in same plane as anterior border of lateral condyle
Type B	Crossing sign, supratrochlear spur	Flat/convex trochlea; spurring about proximal aspect of trochlea
Type C	Crossing sign, double contour	Trochlear facet asymmetry (convex lateral facet, hypoplastic medial facet); anterior border of lateral condyle lies anterior to anterior border of medial condyle
Type D	Crossing sign, double contour, supratrochlear spur	All 3 findings present with characteristic “cliff” pattern (lateral trochlear vertical sloping)

## Data Availability

No new data were created or analyzed in this study. Data sharing is not applicable to this article as this is based on clinical techniques.

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
