# Peer review of "Coronal and Transverse Malalignment in Pediatric Patellofemoral Instability"

_jcm, 2021, doi:10.3390/jcm10143035_

Round 1

Reviewer 1 Report

Coronal and Transverse Malalignment in Pediatric Patellofemoral Instability

Many thanks for the authors having presented this review about coronal and transverse malalignment in pediatric patellofemoral instability. I believe that this study is interesting because it is a complete and well-structured work that ranges from the clinical and radiographic aspects that can cause patellofemoral instability to the possible surgical treatments of skeletal deformities that can cause it.

In some points, mainly in the radiographic evaluation of coronal and transverse rotations, the work is very precise and detailed.

This work represents a great summary of the literature and it contains very useful information for the orthopedic community as guidance in the choice of suitable therapy for this type of pathology.

Comments:

  • In my opinion a description, albeit brief, of the medial patellofemoral instabilities can further improve and complete the work.
  • Page 5 Line 193: The mechanical axis of the lower extremity is determined by a line drawn from the center of femoral head and the center of tibio-talus mortar.

Author Response

Point 1: Many thanks for the authors having presented this review about coronal and transverse malalignment in pediatric patellofemoral instability. I believe that this study is interesting because it is complete and well-structured work that ranges from the clinical and radiographic aspects that can cause patellofemoral instability to the passible surgical treatment of skeletal deformities that can cause it.

In some points, mainly in the radiographic evaluation of coronal and transverse rotations, the work is very precise and detailed.

This work represents a great summary of the literature and it contains very useful information for the orthopedic community as a guidance in the choice of suitable therapy for this type of pathology.

Response 1: We sincerely appreciate the thoughtful feedback on this project.

Point 2: In my opinion a description, albeit brief, of the medial patellofemoral instabilities can further improve and complete the work.

Response 2: Thank you for identifying a very important consideration of medial patellofemoral instability.  As expected, this study focuses exclusively on lateral patellofemoral instability.  Based on your comment, we felt it was important to clarify in the introduction (see line 55-58).

Point 3: Page 5 Line 193: The mechanical axis of the lower extremity is determined by a line drawn from the center of femoral head

Response 3: Please see revised line 195-196 to reflect suggestion

Reviewer 2 Report

I would like to congratulate the authors on this excellent review. I do have some small notes:

1. please add "can" in line 85 - genu valgum "can" be quantified

2. Figure 4 - on this standing radiograph the patellas are not facing forward. In line 209 there is a statement that the patella should face forward on the radiograph. This statement is correct and very important. Therefore, I would recommend having a good representing radiograph with patellas placed exactly in the middle of the knees.

3. line 331 - what "aa" stands for?

4. In figure 10 I suggest adding a mechanical axis line on both radiographs to show the valgus deformity correction.

Author Response

Point 1: Please add “can” in line 85- genu valgum “can” be quantified

Response 1: Please see revised line 87 to reflect suggestion

Point 2: Figure 4- on this standing radiograph, the patellas are not facing forward. In line 209 there is a statement that the patella should face forward on the radiograph. This statement is correct and very important. Therefore, I would recommend having a good representing radiograph with patellas place exactly in the middle of the knees.

Response 2: This comment is appreciated and figure 4 has been updated to reflect suggesting positioning.

Point 3: Line 331- what “aa” stands for?

Response 3: Please see revision of line 334, changing “aa” to “transverse”

Point 4: In figure 10 I suggest adding a mechanical axis line on both radiographs to show the valgus deformity correction.

Response 4: This comment is appreciated and figure 10 has been updated to include a mechanical axis line to both radiographs
